# Water Supply Increases N Acquisition and N Resorption from Old Branches in the Leafless Shrub *Calligonum caput-medusae* at the Taklimakan Desert Margin

Caibian Huang [1,2,3,]*, Fanjiang Zeng [1,2,3], Bo Zhang [1,2,3], Jie Xue [1,2,3] and Shaomin Zhang [4,]*

1    Xinjiang Key Laboratory of Desert Plant Roots Ecology and Vegetation Restoration, Xinjiang Institute of Ecology and Geography, Chinese Academy of Sciences, Urumqi 830011, China; zengfj@ms.xjb.ac.cn (F.Z.); zhangbo@ms.xjb.ac.cn (B.Z.); xuejie11@ms.xjb.ac.cn (J.X.)
2    State Key Laboratory of Desert and Oasis Ecology, Xinjiang Institute of Ecology and Geography, Chinese Academy of Sciences, Urumqi 830011, China
3    Cele National Station of Observation and Research for Desert-Grassland Ecosystems, Cele 848300, China
4    Institute of Nuclear Technology and Biotechnology, Xinjiang Academy of Agricultural Sciences, Urumqi 830091, China
*    Correspondence: huangcaibian@ms.xjb.ac.cn (C.H.); zhangshaomin8698@126.com (S.Z.)

**Abstract:** Irrigation is the main strategy deployed to improve vegetation establishment, but the effects of increasing water availability on N use strategies in desert shrub species have received little attention. Pot experiments with drought-tolerant shrub *Calligonum caput-medusae* supplied with water at five field capacities in the range of 30–85% were conducted using local soil at the southern margin of the Taklimakan Desert. We examined the changes in plant biomass, soil N status, and plant N traits, and addressed the relationships between them in four- and seven-month-old saplings and mature shrubs after 28 months. Results showed that the growth of *C. caput-medusae* was highly responsive to increased soil moisture supply, and strongly depleted the soil available inorganic N pools from 16.7 mg kg$^{-1}$ to an average of 1.9 mg kg$^{-1}$, although the total soil N pool increased in all treatments. Enhancement of biomass production by increasing water supply was closely linked to increasing total plant N pool, N use efficiency (NUE), N resorption efficiency (NRE), and proficiency (NRP) in four-month saplings, but that to total plant N pool, NRE, and NRP after 28 months. The well-watered plants had lower N concentrations in senesced branches compared to their counterparts experiencing the two lowest water inputs. The mature shrubs had higher NRE and NRP than saplings and the world mean levels, suggesting a higher N conservation. Structural equation models showed that NRE was largely controlled by senesced branch N concentrations, and indirectly affected by water supply, whereas NRP was mainly determined by water supply. Our results indicated that increasing water availability increased the total N uptake and N resorption from old branches to satisfy the N requirement of *C. caput-medusae*. The findings lay important groundwork for vegetation establishment in desert ecosystems.

**Keywords:** *Calligonum caput-medusae*; N resorption; water addition; soil inorganic N; biomass

## 1. Introduction

Approximately 10% of drylands undergoes desertification, whereas occurring areas occupy approximately 20% of the dryland population [1]. Establishing vegetation is an important tool for controlling desertification and reducing erosion in desert ecosystems [2–4]. Irrigation is the primary intervention to improve the success of vegetation establishment in desert ecosystems with low precipitation and high evapotranspiration rates [5,6]. For example, shrub planting has been a crucial strategy in the Taklimakan desert highway shelterbelt project, which crosses the largest mobile desert in China and uses drip irrigation to support vegetation to reduce wind-blown sand that blocks the road [7]. In addition to water scarcity,

nitrogen (N) limitation is another primary factor controlling plant growth in desert ecosystems [8], and they generally interact and affect plant growth. Therefore, understanding the N use strategies of artificial vegetation is crucial for the success of vegetation establishment and continuous irrigation programs in extremely arid land.

Soil moisture plays an important role in regulating N mineralization and soil N availability. Reports showed that increasing water availability increased N mineralization rate and N uptake and subsequently promoted plant growth, but reduced moisture and specific soil properties (such as high soil alkalinity), thereby limiting soil N availability [9–11]. Some studies have reported that low N deposition (<6 g $Nm^{-2} \cdot year^{-1}$) improved the plant productivity under drought stress, but high N deposition did not [12,13]. Moreover, increases in soil N availability after N supplementation could improve plant growth and alleviate the negative effects of drought stress in arid land [14]. However, plants became mildly N constrained under sufficient moisture in the desert [15]. The latest research has shown that the annual N deposition was 0.4 g $Nm^{-2} \cdot year^{-1}$ in desert ecosystems of northwest China [16]. However, whether N deposition is a potential approach to mitigate N limitation for the irrigated plants in the desert is uncertain. A number of studies have reported that nutrient resorption, which is the nutrient movement from senescing tissues back to surviving tissues [17], is important especially for plants growing in infertile soils [18–20]. Generally, two approaches can be used to assess nutrient resorption, namely, resorption efficiency and proficiency. Nutrient resorption efficiency quantifies the percent of conserved nutrients in young foliage or other live parts that are translocated from senesced tissues, and resorption proficiency measures the extent to which a nutrient is withdrawn from senescing tissues [19]. Through this nutrient resorption, plants are less dependent on soil nutrient pools to maintain or increase biomass and photosynthesis [20,21]. For example, N resorption in annual plants can provide approximately 31% of N demand [22]. In forests, 45–68% of growth may depend upon resorption [23]. Furthermore, N resorption impacts litter decomposition, nutrient cycling, and resource use efficiency [24–26], thereby affecting plant productivity and nutrient cycling processes. Therefore, clarifying the potential function of NR in artificial vegetation can evaluate their potential fitness and suitability for establishment in infertile and harsh environments such as in the margins of deserts.

The relationship between N resorption and soil fertility is complex given that negative, and positive correlations with soil nutrient availability have been reported [27,28]. The role of water availability on N resorption is also under debate because some studies found that N resorption efficiency (NRE) decreases with increasing soil water availability due to the enhancement of soil nutrient release [29,30]. However, drought can cause early onset of senescence [31], potentially increasing the importance of nutrient resorption [32]. However, nutrient resorption may be sensitive to drought limitation [33,34] due to the reduced nutrient retranslocation in the phloem and water recycling in the xylem [35]. Therefore, whether N resorption in shrubs in arid environments depends on water availability and whether growth stimulation by watering increases N resorption to satisfy increased plant N demand remains unclear.

Seedling establishment of woody species in harsh and extreme environmental conditions is the most vulnerable stage in vegetation establishment [36,37]. Several studies have reported that N limitation at the seedling stage restricted vegetative growth [38], and subsequent vigorous seedling establishment [39]. Moreover, mature trees tend to be more efficient in N-recycling than younger ones [23]. Therefore, N resorption, as an important N recycling strategy, is reported to change over time [40], and its resorption efficiency is likely to increase when plant-available N is limited [21]. Thus, we hypothesize that the seedlings of woody species would have a lower N resorption than adults in a nutrient-poor environment.

We investigated N uptake and utilization including resorption in the drought-tolerant shrub *Calligonum caput-medusae* under different irrigation conditions during seedling establishment in the southern margin of the Taklimakan Desert, Xinjiang, China. Soils in this area are highly weathered and strongly leached, leading to very low nutrient concentrations [41].

*C. caput-medusae* is a perennial C4 plant belonging to Polygonaceae. Its foliage is reduced, making the assimilating green branches the primary photosynthetic organs. It continues to produce new green branches and shed old branches throughout the entire growing season. In addition, the older parts of the green branches of its seedlings become lignified, but more biomasses are allocated to the stem wood as the plant grows. We analyze how seedlings and 28-month-old well-established *C. caput-medusae* plants satisfy their N demand under different water regimes. We compare the effects of water addition on plant growth and N use characteristics to determine the relationships between water availability and N absorption as well as the utilization in seedling and established plants with senescent and green branches of the woody leafless shrub.

## 2. Materials and Methods

### 2.1. Study Area

The study was carried out in Cele Oasis which is located at the southern margin of the Taklimakan Desert in southern Xinjiang, China (Figure 1). The oasis has a typical arid continental climate with an annual mean temperature of 11.9 °C, mean annual precipitation of less than 40 mm (mainly occurring in May and July), and evaporation of approximately 2600 mm. Temperature ranges from 42 °C in summer to −24 °C in winter. The average annual wind speed is 1.9 m·s$^{-1}$, and the maximum speeds in excess of 20.0 m·s$^{-1}$ occur on more than 40 days per year. The frost-free period lasts 209 days per year. The soil is classified as aeolian sandy soil and irrigated desert soil according to the Chinese Soil Taxonomy; they are equivalent to Entisols and Inceptisols in the U.S. Soil Taxonomy, respectively. The Cele Oasis is surrounded by a 5 to 10 km belt of sparse vegetation (5–20% coverage) dominated by *Alhagi sparsifolia*, *Karelinia caspica* and *Tamarix ramosissima*.

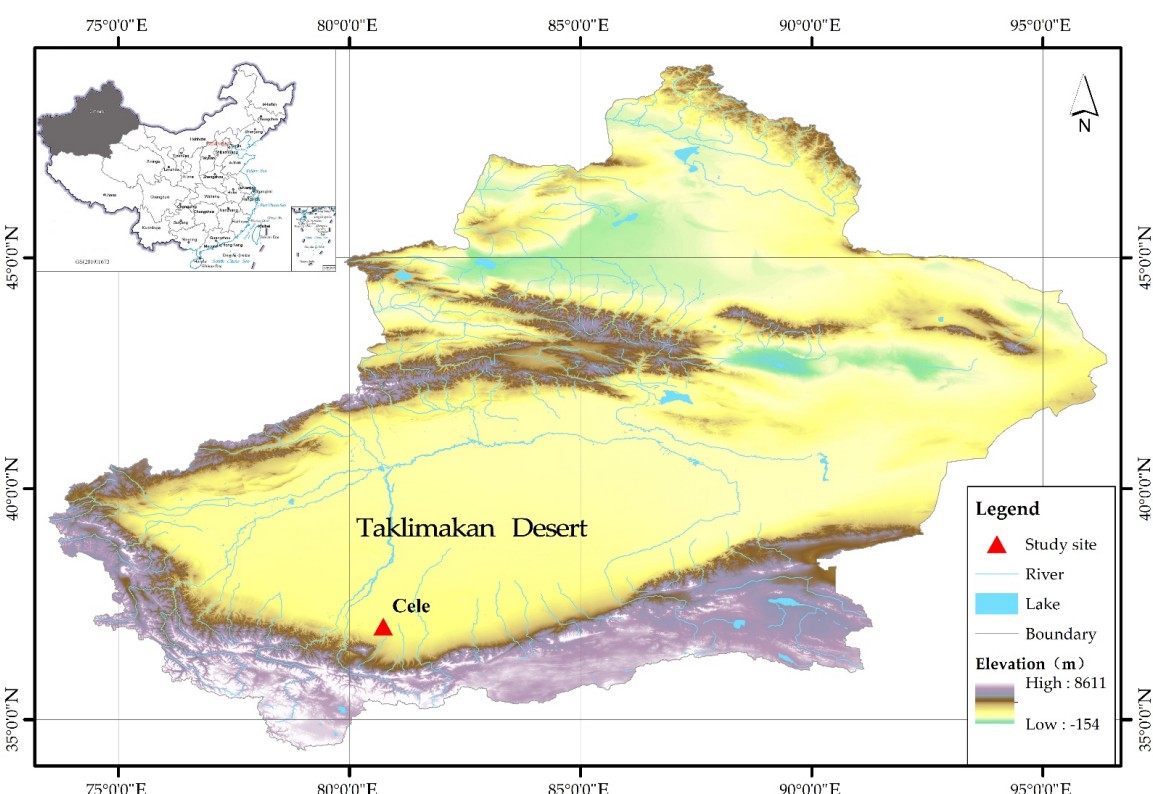

**Figure 1.** Study site at the southern margin of the Taklimakan Desert in southern Xinjiang, China.

### 2.2. Experimental Design

The pot experiment was initiated in April 2011 in an isolated and enclosed natural site to avoid disturbance. The experimental soil (0–40 cm) was collected from an oasis-desert ecotone, was air-dried, and passed through a 2 mm sieve. The characteristics of

the experimental soil are shown in Table 1. A total of 85 kg soil was placed in each of the 150 plastic pots (40 cm inner diameter at the bottom, 50 cm inner diameter at the top edge, and 60 cm tall), to a bulk density of 1.4 g·cm$^{-3}$. The soil field capacity (FC) in the pots was 18%. Seven uniform holes had been drilled in the bottom of each pot and two layers of nylon mesh (0.25 mm) were placed over the holes to prevent root growth out of the pot, but aeration and drainage were. The pots were set in soil to provide thermal buffering with the top edge of each pot extending 3 cm above the ground. A plastic plate was placed on the bottom of each pot to eliminate water transfer into the surrounding soil. Healthy seeds of *C. caput-medusae* with similar sizes were collected in the autumn of 2010, and eight seeds were sown into each pot, and each seed was placed in 2 cm deep holes 5 cm apart on 5 April 2011. For the initial seedling establishment, all pots were well watered (soil moisture was approximately 80% FC) to ensure seed germination. When the one-month seedlings were thinned to one plant per pot, and different water treatments were then initiated. The shoots and roots of the 20 thinned seedlings were harvested and oven-dried at 75 °C, and then weighed to obtain the total biomass of the one-month-old seedlings with 2.4 g plant$^{-1}$ on average.

**Table 1.** Basic soil characteristics before the start of the experiment.

| pH | Organic Matter (g kg$^{-1}$) | Total N (g kg$^{-1}$) | Total P (g kg$^{-1}$) | Total K (g kg$^{-1}$) | Inorganic N (mg kg$^{-1}$) | Available P (mg kg$^{-1}$) | Available K (mg kg$^{-1}$) |
|---|---|---|---|---|---|---|---|
| 8.04 | 2.18 | 0.17 | 0.49 | 14.6 | 16.71 | 1.7 | 145.7 |

The pots were completely randomized and allocated to five water treatments (Figure 2), namely, water-stressed (30% and 40% FC), moderately-watered (50% and 60% FC), and well-watered (85% FC) with 30 replicates for each water level (10 per harvest). During the experimental period, volumetric soil water contents in four randomly selected replicate pots of each water level were measured using a moisture meter type TDR 300 (soil moisture equipment, Santa Barbara, CA, USA) every other day at 20:00. The amount of water to add to each pot was calculated according to the average of four measured pots under each water treatment and enabled the soil volumetric water content to be maintained at 5.4%, 7.2%, 9.0%, 10.8%, and 15.3%, and under 30%, 40%, 50%, 60%, and 85% FC water supply regimes, respectively. The water added to the plants was obtained from the local well without purification. We ignored the N effect from the well water because it contained little nitrate-N (2.92 mg·L$^{-1}$) and no ammonium-N. The water treatments were stopped at the end of the growing season (mid-November) and were re-started at the beginning of the next growing season (mid-April). The experiment was conducted for three years.

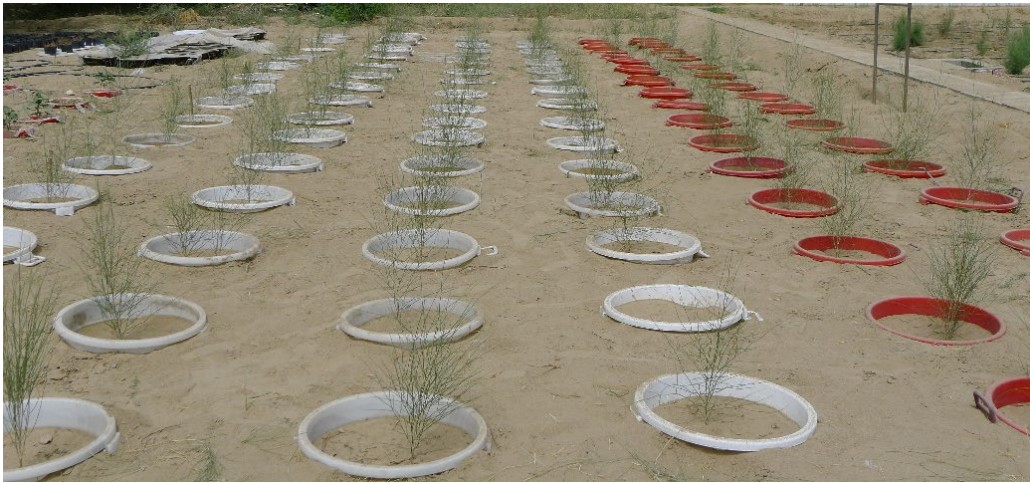

**Figure 2.** Experimental pots and growth status of *C. caput-medusae* after 2 months of water treatment.

### 2.3. Sample Collection

Plants were harvested three times from seedling to mature stages (Table 2). At each harvest time, four individual plants of each water treatment were randomly selected and separated to stem, branches, and roots. For each individual plant, all senesced but still attached branches were collected from each individual and combined into one sample per pot. All plant samples were oven-dried at 75 °C for 48 h and then weighed. Soil samples from 0–15, 15–30, and 30–45 cm depths in each pot were collected after the plant sampling. In each pot, three 2 cm-diameter soil cores were sampled by hand-auger and combined as a single composite sample. All fresh soil samples were sieved through a 2 mm mesh sieve to remove roots and stones and then divided into two subsamples.

**Table 2.** The sampling arrangement and parameters of analysis.

| Sampling Time | Age | Samples | Parameters |
|---|---|---|---|
| Mid-August 2011<br>Late October 2011<br>Mid-August 2013 | 4 months<br>7 months<br>28 months | Green and senesced branches, stems, and roots; soil in each sampling pot | Biomass, total N concentration in each organ, soil total N, nitrate-N ($NO_3^-$), ammonium-N ($NH_4^+$) concentrations |

### 2.4. Laboratory Analysis

All plant samples were ground with a ball mill and then analyzed for the total N concentration by the Kjeldahl acid-digestion method [42] with an Alpkem auto analyzer (Kjeltec System 8400 distilling unit, Foss, Copenhagen, Denmark). The N concentrations were expressed on a mass basis. One soil subsample (10 g) was freshly extracted with 50 mL of 0.01 M CaCl and analyzed for $NO_3^-$ and $NH_4^+$ with a continuous flow analysis system (SEAL Analytical, Norderstedt, Germany). The other soil subsample was air-dried, passed through a 0.25 mm mesh, and then analyzed for soil characteristics. Soil total N concentration was determined by the semimicro Kjeldahl method [43]. Soil pH was determined with a 1:5 soil/water suspension; soil bulk density was measured by the soil core method; soil organic matter was determined by wet oxidation; the total phosphorus (P) was determined after digestion with spectrophotometer detection; soil-available P was extracted with 0.5 M $NaHCO_3$ solution and measured by colorimetric detection; soil total potassium (K) and available K were determined using a flame photometer [44].

### 2.5. Calculation

Individual plant biomass was calculated as the sum of the biomass for each organ (stem, branches, and roots). The total N uptake of each pot was calculated as the sum of individual organ N pools, where individual organ N pool was calculated by multiplying biomass (g·plant$^{-1}$) and its N concentration (mg·g$^{-1}$). The N use efficiency (NUE) was calculated as the ratio of total biomass to total N mass in the whole plant [45]. Nitrogen resorption efficiency (NRE) was calculated as NRE = $(1 - N_{senesced}/N_{green}) \times 100\%$, where $N_{senesced}$ and $N_{green}$ are the N concentrations in senesced and green branches, respectively [19]. We used the reciprocal of $N_{senesced}$ to calculate N resorption proficiency (NRP), where a lower N concentration in senesced tissue corresponds to a higher proficiency [19].

Relative growth rate (RGR) was calculated as the increase in biomass over time: RGR = $(\log_{10}M_f - \log_{10}M_i)/(t_f - t_i)$, where $M_i$ and $M_f$ are the total individual biomass at the end of the fourth month and the first month, the seventh month and the fourth month, and the twenty-eighth month and the seventh month.

### 2.6. Statistical Analysis

One-way analysis of variance (ANOVA) was used to assess the effects of different water treatments on plant biomass, RGR, total N pool, NRE, NUE, NRP, N concentration, and soil N. Means were compared by Duncan's tests where ANOVA showed a significant difference. Two-way ANOVA was used to test the effects of plant age and water addition on soil N and plant N parameters. Regression analyses were used to determine the relationships among plant biomass, soil inorganic N, and plant N traits within each growth

age. All the analyses were carried out with SPSS 16.0 (SPSS Inc., Chicago, IL, USA). We used the package "piecewise structural equation modeling" (piecewise SEM) to analyze the direct and indirect influences of water and plantation age on N concentrations in green and senesced branches, soil inorganic N, and plant biomass with R software (version 4.0.3) [46]. D-separation test of piecewise SEM was used to verify whether the causal model has important links, and $p > 0.05$ indicates the fitness of the model [47].

## 3. Results and Discussion

### 3.1. Soil N status

Average soil inorganic N concentration decreased from 16.71 mg·kg$^{-1}$ to 3.43, 2.55 and 1.9 mg·kg$^{-1}$ after water treatment for four months, seven months, and twenty-eight months, respectively (Figure 3a). Soil inorganic N concentrations under four- and seven-month treatments decreased with increasing water addition and reached the low values at 28 months. However, no relationship was found between water addition rate and soil inorganic N concentration (Table 3). This finding is consistent with several previous findings that soil N availability decreased as juvenile stands begin to mature [23,48]. However, soil total N concentration increased with plantation age from 0.09 g·kg$^{-1}$ (initial value) to an average of 0.15 g·kg$^{-1}$ (28 months) (Figure 3b). Water addition significantly increased soil total N concentration under 50% FC and 60% FC water treatments at 28 months, but no effects were found in four- and seven-month treatments. Therefore, some studies reported that the decline in soil inorganic N may be related to the slow release of available N through litter decomposition, N mineralization, and nitrification [49,50].

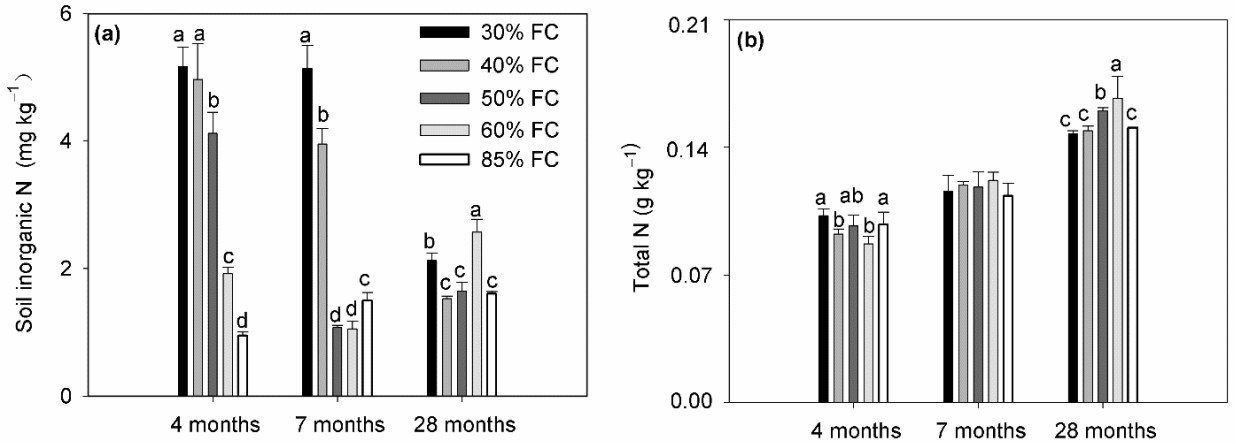

**Figure 3.** Soil inorganic N (**a**) and total N (**b**) concentrations in response to different water treatments at three growth stages (4 months, 7 months and 28months). Values are shown as the means ± se (*n* = 4). Bars with different lowercase letters indicate significant differences among treatments at the same growth stage at $p < 0.05$.

**Table 3.** Pearson's correlation coefficients between water addition rate and measured plant trait and soil N parameters for plants at 4-, 7-, and 28-month age.

| Parameters | Water Addition Rate | | |
|---|---|---|---|
| | **4-Month** | **7-Month** | **28-Month** |
| Soil inorganic N | −0.940 ** | −0.735 ** | −0.113 |
| Soil total N | −0.136 | −0.168 | 0.216 |
| Individual biomass | 0.947 ** | 0.984 ** | 0.995 ** |
| Relative growth rate | 0.964 ** | 0.890 ** | 0.597 ** |
| Total N uptake | 0.736 ** | 0.956 ** | 0.997 ** |
| Green branch N concentration | −0.721 ** | −0.638 ** | 0.362 |
| Senesced branch N concentration | −0.939 ** | −0.515 * | −0.685 ** |
| N resorption efficiency | 0.766 ** | 0.053 | 0.665 ** |
| N use efficiency | 0.491 * | 0.514 * | −0.879 ** |

Notes: * $p < 0.05$; ** $p < 0.01$.

### 3.2. Plant Biomass and Relative Growth Rate

Individual plant biomass and RGR were significantly affected by plant age, water treatments, and their interaction (Table 4). The biomass increased significantly with ages and increasing water supply markedly enhanced this trend at three growth stages (Figure 4a), suggesting that *C. caput-medusae* showed strong adaptability to the decline in soil available N. This finding was consistent with the results in arid and semi-arid areas, indicating that water availability was positively correlated with productivity [30,51]. Increased water availability could directly stimulate plant physiological processes, and consequently increase net carbon uptake [51,52], resulting in high biomass accumulation. Nonetheless, the RGR decreased with plant age, at 0.35, 0.22, and 0.05 for four-, seven-, and 28-month-old saplings, respectively (Figure 4b). The RGR was also improved with an increasing water supply at three growth stages. However, the fourth- and seven-month-old plants grew much faster than the 28-month-old plants.

**Table 4.** Results (*F* value) of two-way ANOVA on the effects of water supply (W), plantation ages (A) and their interactions on soil N, plant biomass and plant N traits.

| Treatments | $N_{inorganic}$ | $N_{total}$ | Biomass | RGR | N pool | $N_{green}$ | $N_{senesced}$ | NRE | NUE |
|---|---|---|---|---|---|---|---|---|---|
| Water (W) | 289.7 *** | 7.11 *** | 1721 *** | 96 *** | 3570 *** | 32.1 *** | 43.2 *** | 22.1 *** | 38 *** |
| Age (A) | 211.2 *** | 233.3 *** | 12,700 *** | 3492 *** | 17,740 *** | 54.9 *** | 168.6 *** | 60 *** | 1161 *** |
| W × A | 117.3 *** | 1.4 | 1427 *** | 19 *** | 2963 *** | 34.5 *** | 15.1 *** | 10.5 *** | 115.3 *** |

Notes: $N_{inorganic}$ and $N_{total}$ correspond to soil inorganic and total N concentrations; RGR and N pool correspond to relative growth rate and whole plant N pool; $N_{green}$ and $N_{senesced}$ correspond to N concentrations in green and senesced branches; NRE, and NUE correspond to N resorption efficiency and N use efficiency; *** $p < 0.001$.

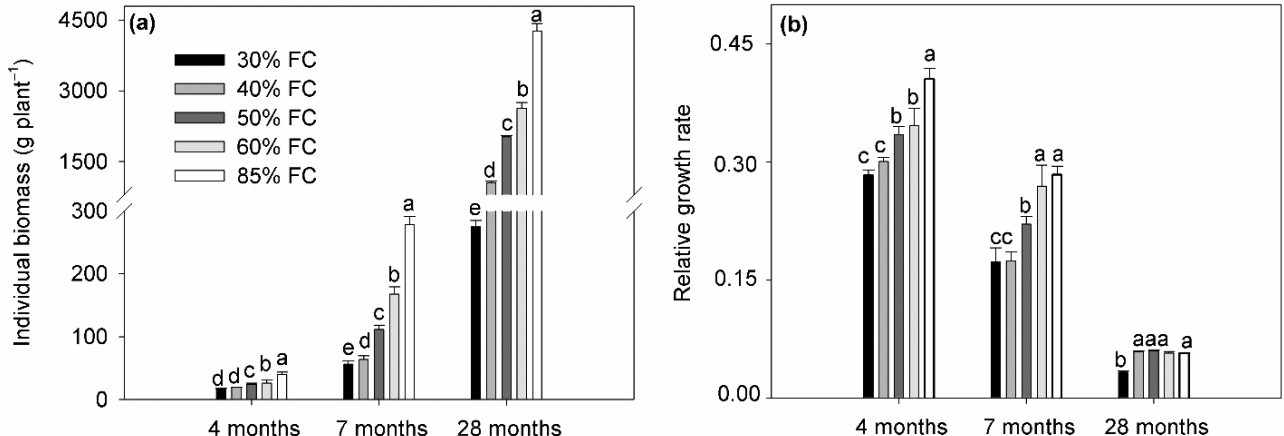

**Figure 4.** Changes in individual plant biomass (**a**) and relative growth rate (**b**) of *Calligonum caput-medusae* under different water treatments at three growth stages (4 months, 7 months, and 28 months). Values are shown as the means ± se (*n* = 4). Bars with different lowercase letters indicate significant differences among treatments at the same growth stage at *p* < 0.05.

### 3.3. Plant N Status

Water addition, plant age, and their interactions had significant effects on N concentrations in green and senesced branches (Table 4, Figure 5b,c). Plant age was the dominant factor in determining N concentrations. Average green branch N concentration among different water treatments decreased from 14.7 mg·g$^{-1}$ at 4 months old to 11.5 mg·g$^{-1}$ at 28 months old; these findings are lower than the average N concentration of terrestrial plant species (18.6 g·kg$^{-1}$) based on a global study [53]. Senesced branch N concentrations also decreased with plant age, with mean values of 5.3, 4.9, and 2.6 mg·g$^{-1}$ at four-, seven- and 28-month-old, respectively. These results were lower than the critical value (7 mg·g$^{-1}$) reported by Killingbeck [19], implying that the senesced branch N was resorbed almost completely which leads to low litterfall N return to the soil. N was the limiting nutrient for

the growth of *C. caput-medusae*, and the limitation even became severe for mature plants, considering our results.

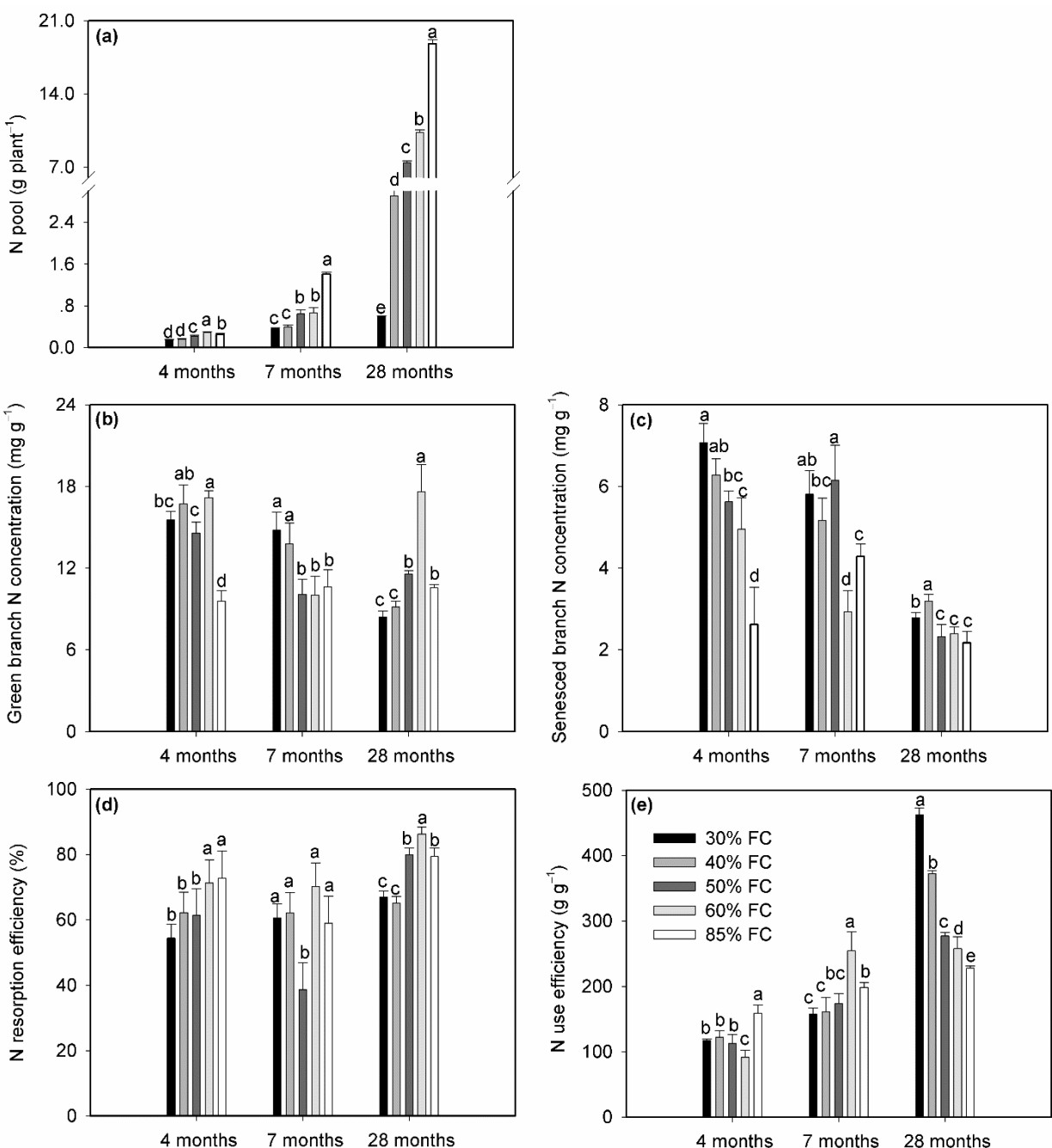

**Figure 5.** Nitrogen pools of total plant (**a**), N concentration in green (**b**) and senesced (**c**) branches, and N resorption efficiency (**d**) and N use efficiency (**e**) of *Calligonum caput-medusae* in response to different water treatments at three growth stages (4-month-old, 7-month-old, and 28-month-old). Values are shown as the means $\pm$ se ($n = 4$). Bars with different lowercase letters indicate significant differences among treatments at the same growth stage at $p < 0.05$.

Effects of water addition on green branch N concentration varied with plant ages. Increasing water addition significantly reduced green branch N concentrations of four- and seven-month-old plants but increased that of the 28-month-old plants. This finding is different from the observation in the shallow-rooted annuals and deep-rooted shrubs in the Gurbantunggut Desert, where water addition showed no effects on their green leaf N concentration [15]. Our study further found that plant N pool increased greatly with water addition rates (Figure 5a), suggesting that enhanced water supply significantly increased

plant N uptake from soil [30]. According to these findings, we speculate that the increase in plant biomass may be higher than that of plant N pool with increasing water addition. This finding leads to the dilution effect of biomass on N content in green branches at the seedling stage, but it was inverse at the mature growth stage. This result was evidenced by the plant N pool at 28-month-age that responded more strongly to water addition than at four- and seven-month-age, showing an increase of 30.6 times at 85% FC water treatment relative to that at 30% FC water treatment. Different from green branches, N concentration in senesced branches was significantly and negatively correlated with water addition rates at all three plant growth ages. On the contrary, the NRP (the reciprocal of senesced branch N concentration) increased with increasing water addition. The decline in senesced branch N concentrations may lead to a decrease in N return to the soil.

### 3.4. N Resorption and Utilization

Non-parasitic plants have mainly two pathways for non-parasitic plants to acquire nutrients for new tissue production, as follows: root uptake from the soil, and mobilizing and withdrawing from old organs. The maintenance of N requirements for *C. caput-medusae* seedlings may be achieved through the pathways at the same time, as evidenced by the sharp decline of soil inorganic N and high NRE. The NRE of four- and seven-month-old plants was lower than that of 28-month-old plants, with mean values of 64.4%, 58.1%, and 75.5% (Figure 5d). The NRE of mature plants was higher than the global mean value (62%) [54]. Sun et al. [23] and Han et al. [55] suggested that plants mainly depended on N resorption with increasing limitation of soil available N. Thus, the mature *C. caput-medusae* may have changed its N acquisition process and depended less on root N uptake.

A significant positive relationship was found between NRE and water addition rates for four- and 28-month-old plants, but not for seven-month-old plants (Table 3), resulting in a significant interaction. In addition, the response of NRP to water addition was more significant than that of NRE, further confirming NRP. Thus, N levels in senesced branches were more sensitive for testing plant internal N cycling [27,56]. This finding suggested that enhanced soil water availability could improve plant's dependence on resorption-derived N due to the increasing limitation of soil available N. N use efficiency was also affected by water addition, plant age, and their interaction (Table 4). The NUE increased with plant age, with mean values of 120.5, 189.0, and 318.2 $g \cdot g^{-1}$ for four-, seven-, and 28-month-old plants, respectively (Figure 5e). Reports showed that plants could use limit N more efficiently with increasing water availability [57,58]. However, our results showed that water addition increased NUE of the four- and seven-month-old plants but decreased that of 28-month-old plants. This finding may be due to the mature plants that changed their N use strategies.

### 3.5. Controlling Factors of Plant Growth and N Utilization as Well as Corelationships between Them

Soil inorganic N concentration was positively correlated with green branch N concentrations at all three plant growth ages, but only related to senesced branch N concentrations at four months (Figure 6). Correspondingly, soil inorganic N was negatively correlated with NRE for four-month-old plants, but this finding was not found at other stages. This result may be related to the calculation of NRE based on the percent changes in green and senesced branch N concentrations. Our SEM result showed that green branch N concentrations were mainly determined by soil inorganic N concentration with indirect regulation by plant age and water addition (Figure 7), whereas soil inorganic N concentration and senesced branch N concentrations were mainly determined by water addition. The NRE was largely determined by senesced branch N concentrations, descendingly by green branch N and soil inorganic N concentrations. This finding suggested that the green branch N concentration could directly reflect the response of soil N availability to water addition during the establishment of *C. caput-medusae*. The plant NRE was not always closely related to the changes in green branch N concentrations and soil N availability. The indirect impact of water addition (via changes in senesced branch N concentrations) is an important driver that changes the N resorption of *C. caput-medusae*.

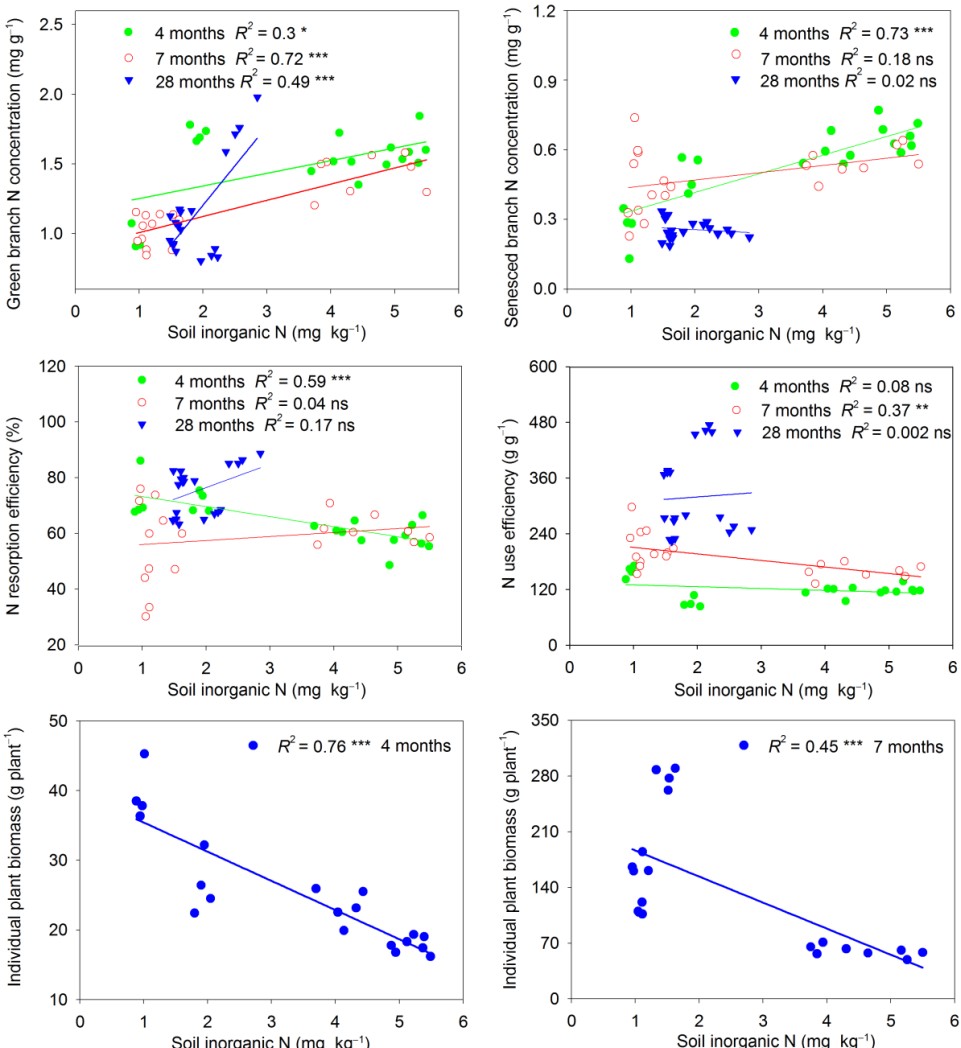

**Figure 6.** Relationships between N concentrations in green and senesced branches, N resorption efficiency, N use efficiency, individual plant biomass, and soil inorganic N. There was no correlation between soil inorganic N and individual plant biomass at 28 months old ($R^2 = 0.01$), and thus not displayed in the plots. Note: * $p < 0.05$; ** $p < 0.01$; *** $p < 0.001$; ns indicates no significant.

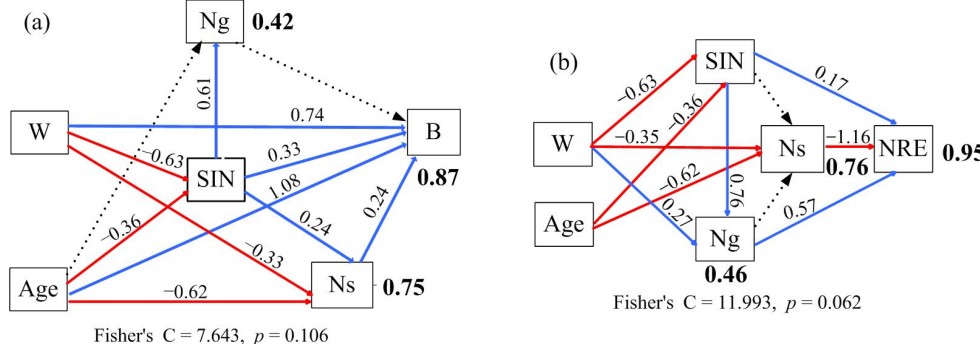

**Figure 7.** Controlling factor analysis of N resorption efficiency and individual plant biomass using the structural equation model. Solid and dashed lines indicate significant ($p < 0.05$) and non-significant ($p > 0.05$) regressions. Blue and red arrows represent positive and negative relationships, respectively. W, water; B, individual plant biomass; SIN, soil inorganic N concentration; Ng, N concentrations in green branches; Ns, N concentrations in senesced branches; NRE, N resorption efficiency. (**a**) controlling factor analysis of individual plant biomass; (**b**) controlling factor analysis of N resorption efficiency.

Individual plant biomass was significantly and negatively correlated with soil inorganic N and green branch N concentrations for the four- and seven-month-old plants, but it was not found for 28-month-old plants (Figure 8). Senesced branch N concentration was strongly and negatively related with individual plant biomass at all three measured times. Therefore, the plant biomass was directly affected by senesced branch N and soil inorganic N concentrations (Figure 7). Our SEM result also showed that water addition played an important direct role in driving plant biomass during the period of seedling establishment of *C. caput-medusae*. The water supply could—both directly and indirectly—regulate plant growth at the same time.

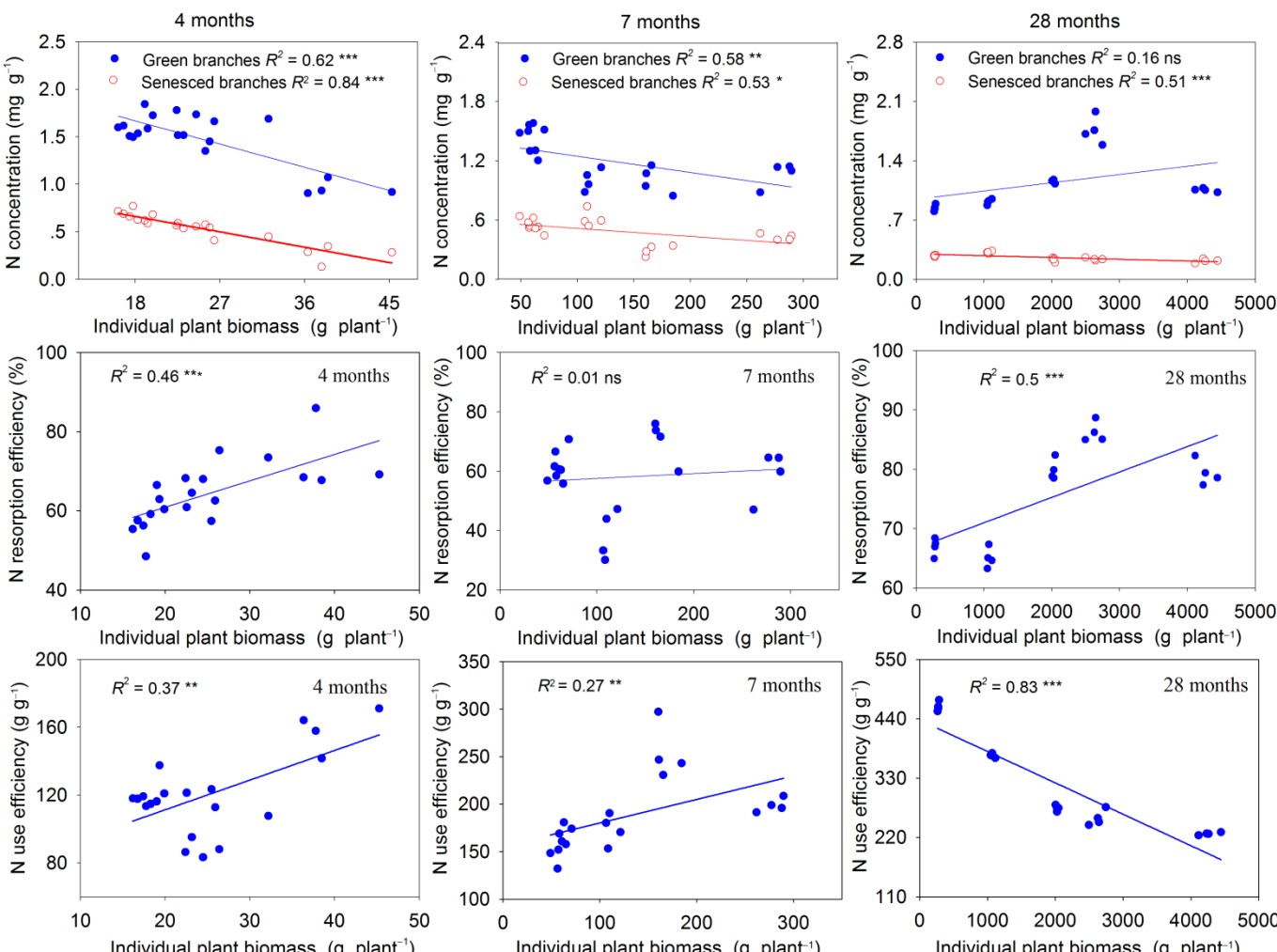

**Figure 8.** Relationships between N concentration in green and senesced branches, N resorption efficiency, N use efficiency, and individual plant biomass of *Calligonum caput-medusae*. Note: * $p < 0.05$; ** $p < 0.01$; *** $p < 0.001$; ns indicates no significant.

Accordingly, we found that plant biomass was positively related to NUE for young plants, but negatively related between them for mature plants (Figure 8). However, plant biomass was positively correlated with NRE at four and 28 months but positively related to NRP at three growth stages. This result indicated that plant biomass and N acquisition were maintained by increasing NUE and N resorption at the seedling stage. Water addition also showed positive effects on these processes. Thus, this result also provided evidence for our finding that plant N acquirement depended on soil N and resorbed N for the saplings. The increasing biomass production of mature plants may be closely related to the high N resorption ability, which may explain the decrease in NUE with water supply. This finding was supported by previous studies, where plants with high productivity had high N resorption [23,59]; thus, their dependence on soil-available N supply was reduced.

## 4. Conclusions

Our results showed that the limitation of soil available N became increasingly serious with plant growth and was exacerbated by water addition. It appears to be an N depletion process for the seedling establishment of *C. caput-medusae*. However, the plants showed strong adaptability to N limitation and satisfied their N requirement by increasing plant N pool, NUE, and N resorption at the seedling stage, but mainly depended on N resorption at the mature stage. Our SEM showed that the individual plant biomass was largely determined by plant age and water addition, and subsequently by soil inorganic N and senesced branch N concentrations which were regulated by plant age and water addition. Enhanced water supply significantly improved plant N uptake from soil and negatively affected soil available N. Water addition mainly promoted NRE by reducing senesced branch N concentrations to maintain plant productivity over the period of seedling establishment of *C. caput-medusae*. Our findings provide a better insight to understand the N adaptive responses to irrigation and lay the groundwork for the vegetation establishment in the hyper-arid ecosystem. Future research is required to explore whether the resorption-derived process can satisfy plant N requirement for a longer time in the study area.

**Author Contributions:** Writing—original draft preparation, C.H.; writing—review and editing, C.H., F.Z. and S.Z.; supervision, S.Z.; project administration, C.H. and F.Z.; methodology, S.Z. and B.Z.; formal analysis, J.X.; software and analysis, S.Z. and J.X. All authors have read and agreed to the published version of the manuscript.

**Funding:** This research was funded by Key Laboratory Project of Xinjiang Uygur Autonomous Region (E0310113), the Original Innovation Project of the Basic Frontier Scientific Research Program, Chinese Academy of Sciences (ZDBS-LY-DQC031), the National Natural Science Foundation of China (42071259), and Natural Science Foundation of Xinjiang Uygur Autonomous Region (2021D01E01).

**Institutional Review Board Statement:** Not applicable.

**Informed Consent Statement:** Not applicable.

**Data Availability Statement:** All data reported here is available from the authors upon request.

**Acknowledgments:** We would like to thank Jonathan R Leake for his helpful suggestions to improve the manuscript.

**Conflicts of Interest:** The authors declare no conflict of interest.

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
