# Peer review of "Water Supply Increases N Acquisition and N Resorption from Old Branches in the Leafless Shrub Calligonum caput-medusae at the Taklimakan Desert Margin"

_water, doi:10.3390/w13223288_

Round 1
Reviewer 1 Report
This manuscript is an interesting work, dealing with a very important scientific topic. The authors are clearly presenting their research results and the manuscript is having a nice structure. I believe this manuscript would be a nice contribution to the Water journal. However, I recommend that the authors should emphasize more and highlight in the introduction part, the contribution of this study regarding the N adaptive responses to irrigation, and state how it would benefit the community. Finally, the authors must make sure that they follow the journal’s format (e.g., the way references are cited)
Author Response
Point 1: I recommend that the authors should emphasize more and highlight in the introduction part, the contribution of this study regarding the N adaptive responses to irrigation, and state how it would benefit the community.
Response 1:
We thank the reviewer’s suggestion. We have added and improved the information in the first and second paragraphs of the introduction to highlight the importance of this study.
The individual Calligonum caput-medusae is often irrigated independently with drip irrigation, and planted with several rows to format shelterbelts. Generally, it is difficult to form a natural plant community in the extreme dry area. In addition, Calligonum caput-medusae grows very fast and cause the obvious natural shadow which is not beneficial for undergrowth. Therefore, we did not talk about the effects of N adaptive responses to irrigation on the community in the introduction, and we mainly focus on the individual plant in our study.
Point 2: The authors must make sure that they follow the journal’s format (e.g., the way references are cited)
Response 2:
Thank you for pointing this out. We have revised the formats of citation and references according to the Journal (Water) style.
Other changes made to the original manuscript:
We separated “Study site”, “Sample Collection” and “Laboratory Analysis” from “Experimental Design” and add two photos about study site and experimental setup. We also merged the results and discussion.
Reviewer 2 Report
- Language needs revision
- Line 18: it is not A pot experiment rather Pot experiments
- line 33: delete the word obvious. There is no obvious in results but conclusions are drawn
- The abstract needs to be a precise account of the findings. Remove redundancy.
- Introduction: What is meant by establishing woody species?
- Line 122 - 124: there is no hypothesis to put ahead when the aim of the study is still being set!!!!
- Materials and Methods: include an atrial photo of the location or site of study.
- Line 143 -147: soil characteristics are to be put in table form for use in the discussion.
- make 2 sections to include in the first study site and the second the experimental design. This section is rather confusing
- Include a photo of the experimental setup so that readers may be able to see and may relate to it.
- Isn't plant and soil sampling part of the experimental design. Use a table to indicate the timeline of your sampling and analysis.
- How much of real chemical analysis was done? this must be put in a separate section.
- Calculations are one entity and statistical analysis is another.
- Merge results and discussion. Start with the soil characteristics and then proceed to the data from the experimental procedure
- The text as it stands needs reorganization.
Author Response
Response to Reviewer 2 Comments
Point 1: Language needs revision
Response 1: The language of our manuscript has been revised by a native English-speaking colleague, and all our co-authors.
Point 2: Line 18: it is not A pot experiment rather Pot experiments
Response2: Thank you for spotting this. We have corrected it to “Pot experiments”.
Point 3: line 33: delete the word obvious. There is no obvious in results but conclusions are drawn. The abstract needs to be a precise account of the findings. Remove redundancy.
Response3: Thanks for the reviewer’s suggestion. We deleted the word “obvious”, and we revised the abstract with precise expression to show our main findings.
Point 4:
Introduction: What is meant by establishing woody species?
Line 122 - 124: there is no hypothesis to put ahead when the aim of the study is still being set!!!!
Response 4: Thanks for pointing this out. We checked the original sentence, and changed “establishing woody species” to “establishing vegetation”. We agreed with the reviewer’s suggestion, and deleted the hypothesis in the last paragraph of the introduction. We also revised the first two paragraphs of introduction according to our findings.
Point 5:
Materials and Methods: include an atrial photo of the location or site of study.
Response 5: We added a photo of our study site as Figure 1.
Point 6:
Line 143 -147: soil characteristics are to be put in table form for use in the discussion.
make 2 sections to include in the first study site and the second the experimental design. This section is rather confusing.
Response 6: We apologize for the confusion caused by classification of this section. We separated study site from experimental design, and put soil characteristics in table form.
Point 7:
Include a photo of the experimental setup so that readers may be able to see and may relate to it.
Isn't plant and soil sampling part of the experimental design. Use a table to indicate the timeline of your sampling and analysis.
Response 7: We added a photo of the experimental setup as Figure 2 in the manuscript. We made a table to indicate the timeline of our sampling and parameters of analysis, and separated this part from the experimental design as Sample Collection.
Point 8:
How much of real chemical analysis was done? this must be put in a separate section.
Calculations are one entity and statistical analysis is another.
Response 8: Thanks again for the suggestion. We put the section of chemical analysis in a separate part, and Calculations in another.
Point 9: Merge results and discussion. Start with the soil characteristics and then proceed to the data from the experimental procedure
The text as it stands needs reorganization.
Response 9: We have merged results and discussion, and fully rewritten this section.